# Targeting of microRNA-22 Suppresses Tumor Spread in a Mouse Model of Triple-Negative Breast Cancer

**DOI:** 10.3390/biomedicines11051470

**Published:** 2023-05-18

**Authors:** Riccardo Panella, Cody A. Cotton, Valerie A. Maymi, Sachem Best, Kelsey E. Berry, Samuel Lee, Felipe Batalini, Ioannis S. Vlachos, John G. Clohessy, Sakari Kauppinen, Pier Paolo Pandolfi

**Affiliations:** 1Cancer Research Institute, Beth Israel Deaconess Cancer Center, Departments of Medicine and Pathology, Beth Israel Deaconess Medical Center, Harvard Medical School, Boston, MA 02215, USA; 2Center for Genomic Medicine, Desert Research Institute, Reno, NV 89512, USA; 3Center for RNA Medicine, Department of Clinical Medicine, Aalborg University, 2450 Copenhagen, Denmark; 4Preclinical Murine Pharmacogenetics Facility and Mouse Hospital, Beth Israel Deaconess Medical Center, Harvard Medical School, Boston, MA 02215, USA; 5Broad Institute of MIT and Harvard, Cambridge, MA 02142, USA; 6Department of Molecular Biotechnology and Health Sciences, Molecular Biotechnology Center, University of Turin, 10154 Turin, Italy; 7Renown Institute for Cancer, Nevada System of Higher Education, Reno, NV 89502, USA

**Keywords:** RNA medicine, microRNA, breast cancer

## Abstract

microRNA-22 (miR-22) is an oncogenic miRNA whose up-regulation promotes epithelial-mesenchymal transition (EMT), tumor invasion, and metastasis in hormone-responsive breast cancer. Here we show that miR-22 plays a key role in triple negative breast cancer (TNBC) by promoting EMT and aggressiveness in 2D and 3D cell models and a mouse xenograft model of human TNBC, respectively. Furthermore, we report that miR-22 inhibition using an LNA-modified antimiR-22 compound is effective in reducing EMT both in vitro and in vivo. Importantly, pharmacologic inhibition of miR-22 suppressed metastatic spread and markedly prolonged survival in mouse xenograft models of metastatic TNBC highlighting the potential of miR-22 silencing as a new therapeutic strategy for the treatment of TNBC.

## 1. Introduction

Breast cancer is the most widely diagnosed cancer in women and is one of the leading causes of cancer-related deaths after lung cancer [1,2,3]. Among the five different breast cancer subtypes, triple negative breast cancer (TNBC) is the most aggressive and lethal subtype [4]. TNBC is characterized by a very strong metastatic potential and its molecular complexity make it particularly hard to treat making the developing for new therapeutic options urgent to address the patient needs. Furthermore, the lack of receptor expression in these subtypes renders them entirely insensitive to endocrine and targeted therapies used for breast cancers [4,5,6]. At present, surgery and chemotherapy are the mainstay of treatment for TNBC patients with a limited number of adjuvant regimens recommended by the National Comprehensive Cancer Network [7]. When surgery with chemotherapy fails, there are no endocrine or targeted therapies available [6]; thus, new treatment strategies for TNBC are urgently needed.

microRNAs (miRNAs) are short non-coding RNAs of about 22 nucleotides in length that function as key post-transcriptional regulators of gene expression by guiding the RNA induced silencing complex (RISC) to partially complementary target sites located predominantly in the 3′ untranslated regions (UTRs) of target mRNAs, resulting in translational repression and/or de-adenylation and degradation of the miRNA targets. Animal miRNAs have been implicated in the regulation of many biological processes, including developmental timing, apoptosis, differentiation, cell proliferation, and metabolism. Furthermore, miRNA dysregulation has been shown to be associated with a wide range of human diseases, such as CNS disorders, cardiovascular and metabolic diseases, and cancer, and thus, miRNAs have emerged as a new class of viable targets for miRNA-based therapeutics [7,8,9,10,11,12,13,14,15,16,17,18,19,20,21,22,23,24,25,26,27,28]. Currently, two strategies are deployed to therapeutically manipulate the activity of disease-associated miRNAs. The biological function of down-regulated or lost miRNAs can be restored using either synthetic double-stranded miRNAs or viral vector-based overexpression, whereas inhibition of overexpressed miRNAs can be achieved using single-stranded, chemically modified antisense oligonucleotides (ASOs) [9,10] called antimiRs. The use of oligonucleotides to manipulate miRNA levels has shown promise in the treatment of a variety of human diseases ranging from inflammation and viral infections to cancer and metabolism [11,12,13,14].

miR-22 is a 22-nucleotide-long microRNA encoded in the exon 2 of the miR-22 host gene (MIR22HG), located on the short arm of Chromosome 17 (GRCh38.p14) in a minimal loss of heterozygosity region. It is highly conserved across many vertebrate species, including chimp, mouse, rat, dog, and horse. This level of conservation suggests functional importance. We have previously shown that miR-22 functions as an oncogene increasing the metastatic potential of breast cancer through its ability to regulate TET2 and the miR-200 family [15,16,17], thereby promoting a strong activation of epithelial-mesenchymal transition (EMT) programs [16]. Specifically, we have previously shown that miR-22 contributes to the metastatic potential of breast cancer in an MCF7 xenograft model and a transgenic mouse model overexpressing miR-22 [15]. In the xenograft models, miR-22 overexpression markedly increased the proliferative rate of MCF7 (Ki67^+^) cells, as well as their metastatic potential to the lung, compared to MCF7 control tumors. Similarly, in transgenic mice engineered to selectively overexpress miR-22 in mammary tissues, we observed a significant decrease in disease-free survival rate resulting from mammary cancer and spontaneous metastatic spread of the disease [15]. These findings prompted us to ask whether miR-22 plays a broader role in breast cancer. Here, we explore the role of miR-22 in TNBC and its impact on EMT and metastatic spread in preclinical models of TNBC.

## 2. Methods

### 2.1. Cell Culture

MDA-MB-231 (ATCC, ATCC HTB-26) cells were either transduced with lentivirus containing a plasmid designed to overexpress miR-22 and express RFP or a plasmid that only expressed RFP. Cells were sorted based on the presence of RFP. RFP-positive cells were collected and grown in high glucose DMEM (Thermo Fisher) containing 500 U/mL of penicillin-streptomycin (Thermo Fisher) and 20 mM L-glutamine (Thermo Fisher). 

### 2.2. Mammospheres Preparation

MDA-MB-2312 cells overexpressing miR-22 and empty vector cells were seeded at a concentration of 4 × 10^3^ cells/cm^2^ in Mammocult media (StemCell Technologies) containing the Mammocult Proliferation Supplement (StemCell Technologies), 4 µg/mL heparin (StemCell Technologies), 0.48 µg/mL hydrocortisone (StemCell Technologies), and 500 U/mL Penicillin-Streptomycin (Thermo Fisher). Mammospheres grew in this media for 16 days, and the medium was changed every 2 days. Mammospheres were given 5 µL/mL doses of anti-miR-22 LNA, SCR-LNA for a final concentration of 500 nM, and VHL every 2 days when the medium was changed.

### 2.3. Synthesis of LNA Oligonucleotides

LNA oligonucleotides were obtained from Exiqon and resuspended in 0.9% NaCl to a concentration of 1 mg/mL in a sterile environment, and filter sterilized using a 0.22 µm vacuum filter. 

### 2.4. RNA Extraction and Purification

At specific time points, cells/mammospheres were spun down at 1200 RPM for 5 min. The supernatant was aspirated; the cell/mammosphere pellet was resuspended in DPBS (Thermo Fisher), and the resuspended cells were spun down at 1200 RPM. The DPBS was aspirated, and 1 mL of TRIzol (Thermo Fisher) was used to lyse the pellet. The lysate was processed using the PureLink RNA Mini Kit (Thermo Fisher) using the instructions provided in the kit. 

### 2.5. RT-qPCR

The purified RNA was subjected to Poly (A) tailing by using *E. coli* Poly (A) Polymerase and rATP from New England BioLabs following the manufacturer’s instructions. Ten µL of the poly (A) tailed-RNA was reverse-transcribed using the SuperScript™ IV First-Strand Synthesis System (Thermo Fisher) following the manufacturer’s instructions. PowerUp™ SYBR™ Green Master Mix (Thermo Fisher) was used in all qPCR reactions (Table 1). 

### 2.6. Protein Extraction and Quantification

RIPA buffer was prepared by adding 20 µL of 50× EDTA-Free protease inhibitor cocktail (Sigma-Aldrich), and 100 µL PhosStop phosphatase inhibitor (Sigma-Aldrich) to 880 µL RIPA buffer (Boston Bioproducts). Mammospheres were pelleted using the same procedure described in the “RNA Extraction and Purification” section, and RIPA buffer plus protease and phosphatase inhibitor were used to resuspend and lyse the pellet. After incubating the cell lysate on ice for 30 min, the lysate was spun at 15,000× *g* for 20 min; supernatants were collected, and the pellet discarded. The protein in the supernatant was quantified using the Bio-Rad Protein Assay Reagent and NanoDrop One^©^ (Thermo Fisher).

### 2.7. Western Blot

3 to 5 µg of protein from each sample was loaded onto a 4–12% Bis-Tris NuPage Gel (Thermo Fisher), and after separation was complete, transferred onto a nitrocellulose membrane and immunoblotted for the proteins of interest. 

### 2.8. Antibodies

The following antibodies were used from immunohistochemistry: anti-Ki67 (1:200, MA5-14520) from Invitrogen, and anti-AXL (1:2000, ab219651) from Abcam. The secondary antibodies used for immunohistochemistry staining were: Amersham ECL Mouse IgG (NA931-1ML) and Amersham ECL Rabbit IgG (NA9340-1mL) antibodies were used (GE Healthcare Life Sciences). 

### 2.9. Immunohistochemistry

Immunohistochemistry staining was performed on the lungs and livers of the xenografted mice being studied. Slides were deparaffinized at 62 °C for 15 min, and then subjected to washes of xylene, 100% ethanol, and 95% ethanol. Antigen retrieval was performed using either a sodium citrate buffer or EDTA buffer based on the primary antibody being used, followed by chemical blocking using 30% hydrogen peroxide (Sigma-Aldrich), serum blocking with rabbit serum (MP Biomedicals), and an overnight incubation at 4 °C with the primary antibody was completed. The following day, the slides were washed with DPBS (Thermo Fisher), and then incubated in the secondary antibody at room temperature for 30 min. This was followed by an incubation in the working solution from the Vectastain ABC-HRP kit from Vector Labs for 30 min, diaminobenzidine (Sigma-Aldrich) staining, and counterstaining with hematoxylin (Fisher Scientific).

### 2.10. Hematoxylin and Eosin Staining

H&E staining was performed on the same samples as above. Slides were deparaffinized using multiple washes of xylene, 100% ethanol, and 95% ethanol. Slides were incubated in hematoxylin (Fisher Scientific), washed with 70% ethanol with 1% hydrochloric acid, followed by an incubation in eosin (Fisher Scientific). After final washes of 95% and 100% ethanol, the slides were ready to have a coverslip. 

### 2.11. Mice

Animal experiments were performed in accordance with the guidelines of Beth Israel Deaconess Medical Center Institutional Animal Care and Use Committee. Nu/J mice were obtained from the Jackson Laboratory, and 5 × 10^6^ of MDA-MB-231 overexpressing miR-22 cells were injected into 30 mice via tail vein, while MDA-MB-231 empty vector cells were injected into other 30 mice using the same delivery method. Tail vein injection was chose as transplant method to better mimic the process of metastasis formation and have the cells in the blood stream and to avoid forcing cell housing in a specific tissue of our choice. Each mouse within the two cohorts was separated into one of three treatment groups, for a total of 10 mice per experimental group, consisting of IP injections of either anti-miR-22 LNA, SCR-LNA, or VHL, given at a dose of 10 mg/Kg. Mice within ±2.5 g of each other were given the same dosage volume. 

### 2.12. Statistical Analysis

All the statistical analysis on qPCR data, staining, and animals was performed with the ANOVA tool (2-way ANOVA or 3-way ANOVA depending on the experiment) included in the Prism Graph Pad 8. P-values are represented as non-significant (ns) for *p* > 0.1; (*) for *p* < 0.03; (**) for *p* < 0.002; (***) for *p* < 0.0002 and (****) for *p* < 0.0001

### 2.13. Bioinformatic Analysis

The miRNA expression data from RNAseq (Illumina Hiseq) of samples with breast invasive carcinoma were downloaded from TCGA on 2 May 2020, from “https://tcga.xenahubs.net/download/TCGA.BRCA.sampleMap/miRNA_HiSeq_gene.gz”. Comprehensive metadata were downloaded from https://tcga.xenahubs.net/download/TCGA.BRCA.sampleMap/miRNA_HiSeq_gene.json on 2 May 2020. Tumors with human epidermal growth factor receptor (HER2) were defined as HER2-positive (HER2+). Tumors that were HER2-negative (HER2-) but were positive for either estrogen receptor (ER) or progesterone receptor (PR) were defined as hormone receptor positive (HR+). Tumors that were negative for ER, PR, and HER2, were defined as triple-negative breast cancer (TNBC). Kruskal–Wallis rank sum test was used to detect differences of miR-22-3p expression levels across all breast cancer subtypes. For pair-wise comparisons, we used Wilcoxon rank sum test with continuity correction (Mann–Whitney U test). Survival analysis was performed using the Kaplan–Meier method. All analyses were done in R (version 3.6.1). Statistical analysis and graphing of all non-bioinformatical data were performed using GraphPad Prism 8.4.2.

## 3. Results

Molecularly, TNBC and HER2+ breast cancers share many similarities as both are enriched in loss-of-function mutations in TP53, PTEN, and RB1. HER2+ breast cancers are further characterized by *ERBB2* amplification. These features result in a higher histological grade and a more aggressive disease. In contrast, hormone-receptor positive (HR+) breast cancers typically share a luminal transcriptional profile, which is often estrogen-dependent and enriched in GATA3 mutations with lower mutation rates of TP53 [18]. In terms of prognosis and life expectancy, breast cancer is divided into four different stages, from I to IV, with stage IV being characterized by metastatic spread and considered incurable as per today. The vast majority of stage IV breast cancers are also TNBC [19]. Our in silico analyses suggest that miR-22 levels are elevated in stage IV breast cancer (Figure 1A) and this observation further prompted us to explore the role of miR-22 in this breast cancer subtype.

We first generated a cellular model from a human TNBC cell line, MDA-MB-231, in which we stably overexpressed miR-22 (Figure 2A). To achieve miR-22 overexpression in human TNBC cell lines, a plasmid containing an RFP reporter, designed to constitutively express miR-22, was transduced into MDA-MB-231 cells (referred to as LV-RFP miR-22). This line was compared to MDA-MB-231 cells transduced with an empty vector (referred to as LV-RFP Empty) containing the same reporter. We then isolated RFP-positive cells by sorting the top 10% of RFP-fluorescent cells. After cell recovery, equal numbers of each cell type were seeded into 10 cm dishes to determine whether overexpression of miR-22 resulted in increased proliferation compared to cells expressing basal levels of miR-22. As shown in Figure 1B, cells overexpressing miR-22 displayed a significant increase in the growth rate over control cells. Next, we asked whether miR-22 overexpression affects EMT in cultured MDA-MB-231 cells. To this end, we extracted RNA from both LV-RFP miR-22 and LV-RFP Empty cell lines and compared the expression levels of key EMT-related genes and mesenchymal markers such as SNAIL1, CDH2 (N-Cadherin), and TWIST1, as well as common epithelial structure markers such as DSP (Desmoplakin), and CDH1 (E-Cadherin). As shown in Figure 1C, we observed a significant increase in EMT-related genes and mesenchymal markers SNAIL1, TWIST1, and CDH2, as well as a significant decrease in the epithelial structural markers DSP and CDH1, implying that miR-22 impacts the EMT signature. To further assess the biological consequences of the aforementioned molecular perturbations, we established 3D cultures of the LV-RFP miR-22 and LV-RFP Empty cell lines by seeding the cells in a specialized medium under conditions (see Methods and Materials for details) that triggered the formation of mammospheres. The experiment was designed to compare the size and number of LV-RFP miR-22 and LV-RFP Empty mammospheres by seeding equal numbers of cells in ultra-low attachment dishes containing the specialized medium mentioned above. As shown in Figure 1D,E and Figure 2B, we observed a significant increase in the size and number of mammospheres in LV-RFP miR-22- seeded dishes compared to LV-RFP Empty controls, confirming the ability of miR-22 to confer stemness properties to human TNBC cells. To further corroborate the relevance of miR-22 in triggering EMT, we extracted RNA from both LV-RFP miR-22 and LV-RFP Empty cells and analyzed by qPCR the expression levels of the same key EMT-related genes as well as the mesenchymal and epithelial related markers listed above. The expression data demonstrate that the effect of miR-22 overexpression on EMT is maintained in mammospheres (Figure 1F).

To investigate whether inhibition of miR-22 function could revert the EMT signature and reduce stemness of cells in mammospheres, we seeded LV-RFP miR-22 cells or LV-RFP Empty cells in specialized medium in the presence of an antimiR-22 oligonucleotide (miR-22 LNA), VHL, or SCR-LNA control, respectively. As shown in Figure 3A–D, by day 8, inhibition of miR-22 reduced the size of mammospheres in both LV-RFP miR-22 and LV-RFP Empty cells, while the total number of mammospheres was only reduced in the LV-RFP Empty samples, probably due to the fact that the inhibition of miR-22 in a cell line not overexpressing the miRNA has faster kinetics. As shown in Figure 4B through Figure 4E, by day 12, inhibition of miR-22 significantly slowed the growth and reduced the numbers of both LV-RFP miR-22 and LV-RFP Empty mammospheres, implying that the antimiR-22 treatment is effective at reducing the stemness of human TNBC cells exhibiting either overexpression or basal levels of miR-22. Indeed, the effect of miR-22 inhibition in mammosphere size could be easily observed under a brightfield microscope as shown in Figure 4F,G. Next, we determined whether the antimiR-22 treatment could revert the genetic signature of EMT in mammospheres. To this end, we obtained RNA from mammospheres treated as shown in Figure 4A. As shown in Figure 4H,I, we observed a partial reversion of EMT in human TNBC cells displaying basal levels of miR-22, and complete reversion of EMT in human TNBC cells overexpressing miR-22. Given the positive findings shown in Figure 1E, we decided to expand the panel of EMT genes. The data confirmed that inhibition of miR-22 function can revert EMT in a relevant 3D model of human TNBC and could represent a potential therapeutic strategy against the progression and metastatic spread of TNBC. 

Finally, we investigated the effect of miR-22 overexpression on the metastatic potential and overall survival of human TNBC xenografts. As shown in Figure 5A,B, mice engrafted with human TNBC overexpressing miR-22 displayed a profound decrease in overall survival. This was caused by a much-increased metastatic spread of tumors to the lung and proliferation of the metastatic nodules as shown by H&E and IHC staining (Figure 5B), indicating that miR-22 overexpression impacts TNBC progression and outcome. We next assessed the in vivo efficacy of the antimiR-22 compound as shown in Figure 5C. Experimental cohorts were treated with either VHL, SCR-LNA, or anti-miR-22 LNA once weekly. Two weeks into the experiment, a postmortem census of both cohorts revealed that pharmacologic inhibition of miR-22 suppressed the metastatic spread and growth of tumors in both xenograft models as shown by H&E and IHC staining (Figure 5D,E). By the end of the experiment, the antimiR-22 treatment had significantly extended survival in both LV-RFP miR-22 and LV-RFP Empty mouse cohorts, as shown in Figure 5F,G. Collectively, these findings suggest that antimiR-22-mediated inhibition of miR-22 may constitute a promising therapeutic strategy for treatment of breast cancer of various molecular subtypes, irrespective of miR-22 expression levels.

## 4. Discussion

Although diagnosis of breast cancer cases has risen over the past decade, death rates have declined, suggesting that currently available therapeutics are effective in extending the lives of those diagnosed. Indeed, data from the NIH SEER database suggest that women diagnosed with local and regional breast cancers display 5-year survival rates of 99% and 86%, respectively [3,20]. However, once breast cancer progresses to present with distant metastatic spread, the outlook changes drastically and the prognosis becomes poor. By Stage III, breast cancer has spread to the lymph nodes surrounding the sternum, clavicles, and axillae, and the overall 5-year survival rate falls to 72%. Once breast cancer progresses to Stage IV and distant metastasis can be found in the lungs, liver, brain, and bones, the overall 5-year survival rate drops to 22%; a clear indication that a great deal of work is still to be done to make breast cancer fully curable [7,20,21,22]. 

Currently available treatments for TNBC are limited to standard chemotherapy, Poly ADP Ribose Polymerase (PARP) inhibitors for tumors harboring BRCA1/2 mutations, and PD-L1 checkpoint blockade for tumors with infiltrating lymphocytes, or tumors that are PD-L1 positive. The development of novel therapeutic strategies is therefore critically important to improve the survival rates of patients with breast cancer [23,24,25]. 

New promising therapeutic strategies have emerged over the past few years that focus on targeting of non-coding RNAs such as miRNAs and long noncoding RNAs (lncRNAs). For example, inhibition of the lncRNA MALAT1 triggered differentiation of breast cancer in animal models, suppressing tumorigenesis [26]. In contrast, MALAT1 overexpression was shown to repress the metastatic potential of TNBC cell lines and was proposed as a strategy for a novel anti-metastatic therapy [27]. These opposing functions of MALAT1, and potentially other ncRNAs, highlight the complexity of ncRNA biology. Hence, the development of new therapeutic approaches based on miRNAs and lncRNAs requires a deep understanding on the biological mechanisms of the targeted ncRNAs. We have previously shown that miR-22 contributes to the metastatic potential of breast cancer in an MCF7 xenograft model and a transgenic mouse model overexpressing miR-22 [15]. In the xenograft models, miR-22 overexpression markedly increased the proliferative rate of MCF7 (Ki67^+^) cells, as well their metastatic potential to the lung, compared to MCF7 control tumors. Similarly, in transgenic mice engineered to selectively overexpress miR-22 in mammary tissues, we observed a significant decrease in disease-free survival rate resulting from mammary cancer and spontaneous metastatic spread of the disease [15]. These findings prompted us to ask whether miR-22 plays a broader role in breast cancer. 

The well-known interaction of miR-22 with TET2 and the subsequent effect on miR-200 family, suggests that miR-22 is a central player in EMT. Through EMT, cancer cells acquire mesenchymal-like traits, and become more invasive, ultimately increasing their metastatic potential. Furthermore, remodeling of the extracellular matrix (ECM) by secretion of proteases allows tumor cells to move freely to their surroundings [28,29]. As tumor invasion progresses, cancer cells intravasate into the lymphatic and circulatory systems, granting them access to distant parts of the body and creating metastatic sites in various organs capable of sustaining tumor cell growth [30]. Thus, it is tempting to speculate that blocking EMT would slow the metastatic spread of cancer. We have previously demonstrated a correlation between metastatic potential of cancer and lipogenic switch [31], and more recently we showed that miR-22 acts as a positive master regulator of lipid metabolism and a key player in metabolic control (Panella et al. submitted). These findings provide further evidence of the crucial role of miR-22 in different and essential aspects of the metastatic process.

In conclusion, we demonstrated that overexpression of miR-22 triggers a more aggressive and mesenchymal-like phenotype in a TNBC model compared to cells expressing physiologic levels of miR-22. We observed a significant increase in the growth rate of the miR-22-overexpressing TNBC cells, increased expression of notable mesenchymal genes, and decreased expression of critical epithelial markers. Successful elucidation of the function of miR-22 in cultured cells and in vivo offers hope for an important and greatly needed treatment for TNBC. Interestingly, antimiR-22 treatment not only impacted the growth of TNBC cells overexpressing miR-22, but was also effective at slowing the growth of TNBC cells expressing basal amounts of miR-22. This finding suggests that an antimiR-22-based therapy could be effective in breast cancer patients irrespective of whether they overexpress miR-22 or exhibit normal expression levels of this miRNA. Additionally, overexpression of miR-22 dramatically reduced survival of TNBC xenograft mice. Given that the mature miR-22 sequence is 100% conserved between mouse and man, our data in mouse models of TNBC are indicative that an antimiR-22-based therapy could be further pursued in TNBC patients.

## Figures and Tables

**Figure 1 biomedicines-11-01470-f001:**
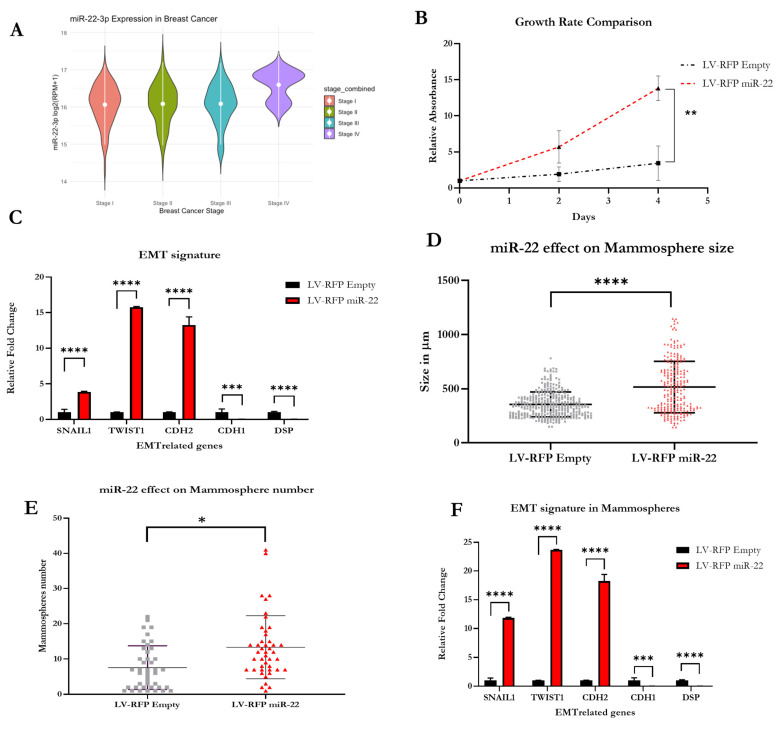
**Overexpression of miR-22 promotes EMT in a 2D model of TNBC.** Bioinformatic analysis of miR-22 expression levels at different stages of breast cancer (**A**). Overexpression of miR-22 causes a significant increase in the growth rate of a 2D human TNBC in-vitro model (**B**). Overexpression of miR-22 results in a significant upregulation of genes related to EMT, as well as downregulation of genetic markers related to genes found in epithelial cells (**C**) Overexpression of miR-22 in a 3D model of TNBC causes a significant increase in the size and number of mammospheres (**D**,**E**). Overexpression of miR-22 results in the significant increase in genes associated with mesenchymal cells, and a significant decrease in genes associated with epithelial cells in a 3D human TNBC model (**F**) [Statistical analysis are represented as (*) for *p* < 0.03; (**) for *p* < 0.002; (***) for *p* < 0.0002 and (****) for *p* < 0.0001].

**Figure 2 biomedicines-11-01470-f002:**
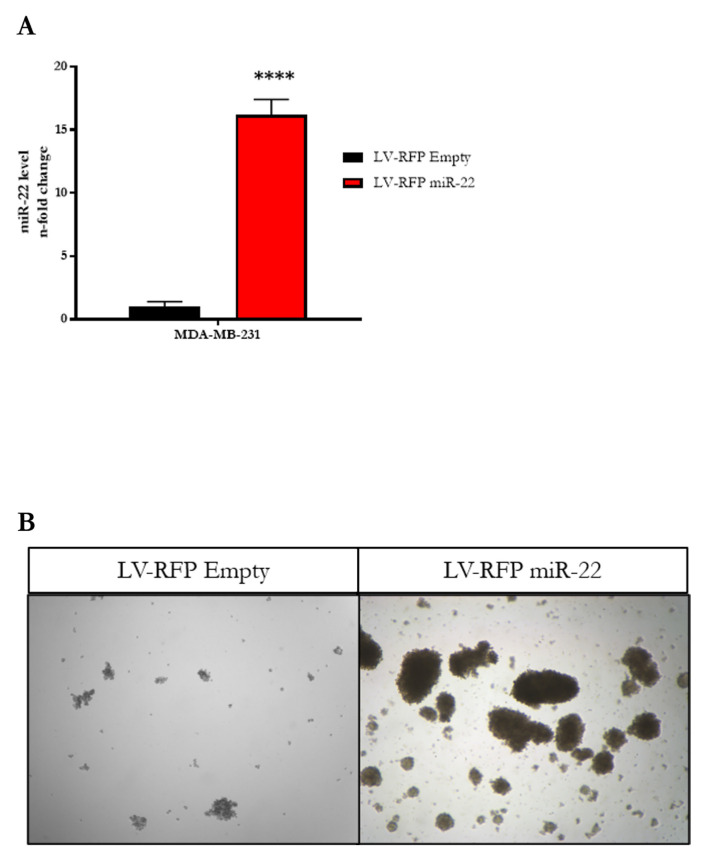
Validation that designed plasmids can upregulate the expression of miR-22 (in the case of LV-RFP miR-22) and maintain basal levels of miR-22 (in the case of LV-RFP Empty) (**A**). Brightfield images of MDA-MB-231 cultured in mammosphere medium in which overexpression of miR-22 results in an increase in the number and size of mammospheres (**B**). [Statistical analysis are represented as (****) for *p* < 0.0001].

**Figure 3 biomedicines-11-01470-f003:**
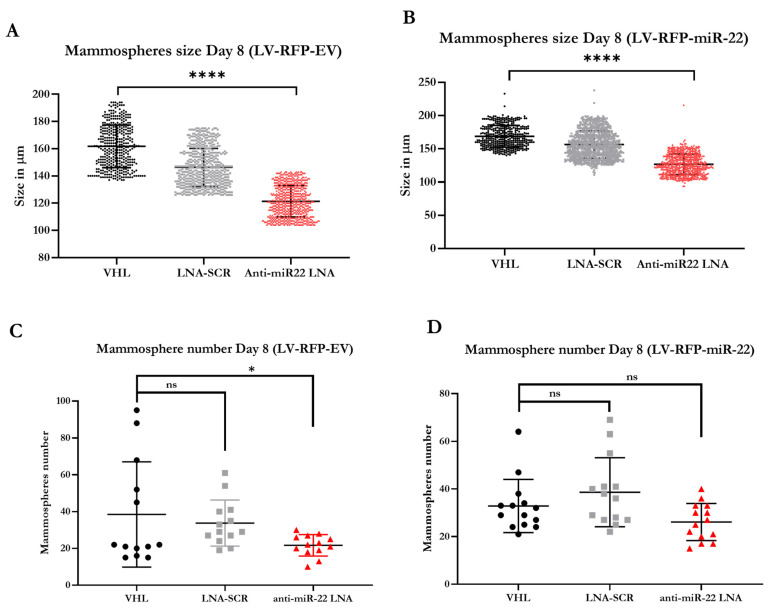
The antimiR-22 treatment is effective at decreasing the size of both LV-RFP miR-22 and LV-RFP Empty mammospheres by day 8, but is only able to significantly reduce the number of LV-RFP Empty mammospheres (**A**–**D**). [Statistical analysis are represented as non-significant (ns) for *p* > 0.1; (*) for *p* < 0.03 and (****) for *p* < 0.0001].

**Figure 4 biomedicines-11-01470-f004:**
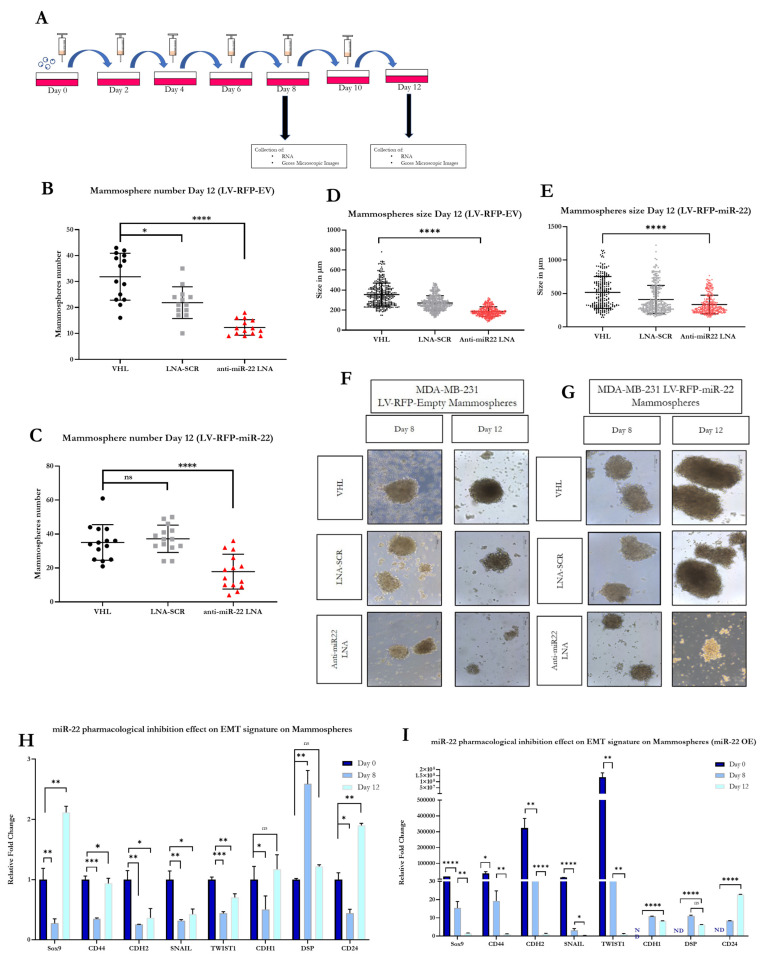
**Inhibition of miR-22 function in TNBC mammospheres.** A schematic of the in vitro experiments with the antimiR-22 oligonucleotide (**A**). Treatment of TNBC mammospheres with antimiR-22 significantly decreases the size and number of LV-RFP miR-22 and LV-RFP Empty mammospheres (**B**–**E**). Visual representations of the effect of antimiR-22 treatment on LV-RFP miR-22 and LV-RFP Empty mammospheres at days 8 and 12, respectively (**F**,**G**). Treatment with antimiR-22 LNA can partially revert the EMT signature of LV-RFP Empty mammospheres, and completely revert the EMT signature in LV-RFP miR-22 mammospheres (**H**,**I**). [Statistical analysis are represented as non-significant (ns) for *p* > 0.1; (*) for *p* < 0.03; (**) for *p* < 0.002; (***) for *p* < 0.0002 and (****) for *p* < 0.0001].

**Figure 5 biomedicines-11-01470-f005:**
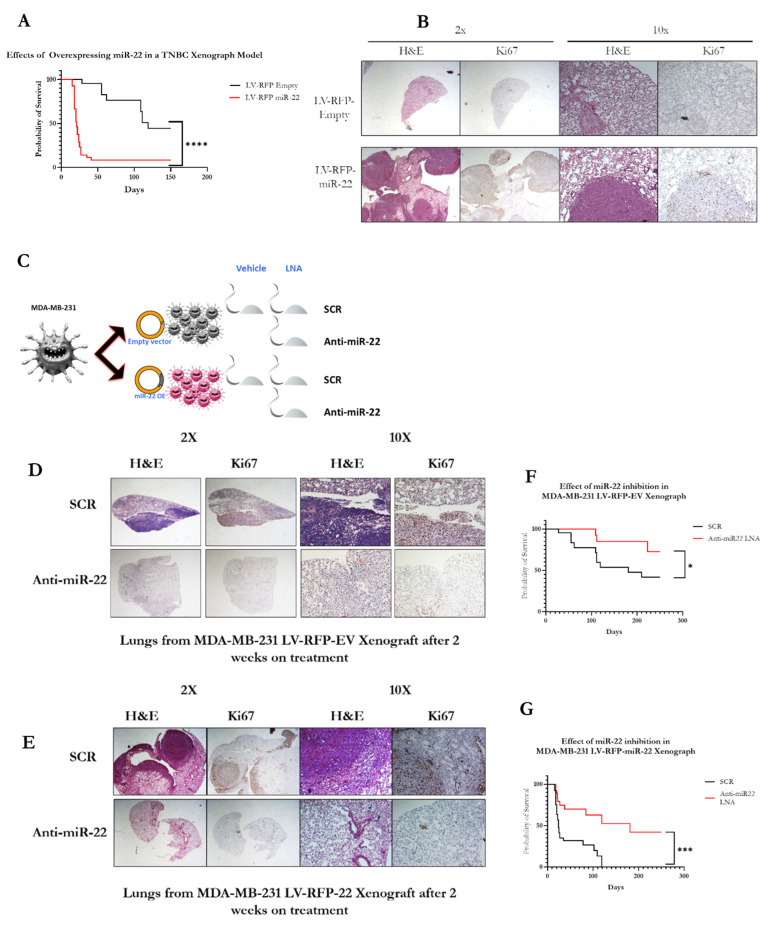
**Therapeutic inhibition of miR-22 in TNBC xenograft mice.** Mouse xenografts overexpressing miR-22 show a significant decrease in life expectancy compared to LV-RFP Empty xenografts, as well as larger tumors as shown by H&E and IHC (**A**,**B**). The animals were separated into three cohorts and treated with either vehicle (VHL), scramble control ASO (SCR), or the antimiR-22 compound (**C**). After two weeks, a census was conducted on the mice, and H&E showed smaller tumor sizes and a decrease in proliferation in the LV-RFP miR-22 and LV-RFP Empty xenograft cells treated with the antimiR-22 compound compared to mice treated with SCR LNA (**D**,**E**). Treatment with antimiR-22 resulted in a significant increase in the survival of LV-RFP miR-22 and LV-RFP Empty mice compared to treatment with vehicle or scramble control-treated mice (**F**,**G**). [Statistical analysis are represented as (*) for *p* < 0.03; (***) for *p* < 0.0002 and (****) for *p* < 0.0001].

**Table 1 biomedicines-11-01470-t001:** Primers for RT-qPCR.

Gene ID	Fwd 5′-3′	Rev 5′-3′
CDH1	tgctcttccaggaacctctg	gcggcattgtaggtgttc
CDH2	cctgaagccaaccttaactga	tggagggatgacccagtct
SOX9	tacccgcacttgcacaac	tctcgctctcgttcagaagtc
DSP	aaagaaaatgctgcctactttca	ggggtacttcttcctgatgga
TWIST	gggccggagacctagatg	tttccaagaaaatctttggcata
SNAIL	gcgagctgcaggactctaat	cggtggggttgaggatct
CD44	tgacacatattgcttcaatgctt	tgggcaggtctgtgactg
CD24	atgggcagagcaatggtg	ccagttgttgtttcactggaat
HUPO	gcttcctggagggtgtcc	ggactcgtttgtacccgttg

## Data Availability

The data presented in this study are available on request from the corresponding author. The data are not publicly available due to patent pending issues.

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
