# Peer review of "Targeting of microRNA-22 Suppresses Tumor Spread in a Mouse Model of Triple-Negative Breast Cancer"

_biomedicines, 2023, doi:10.3390/biomedicines11051470_

Round 1
Reviewer 1 Report
The article “Targeting of microRNA-22 suppresses tumor spread in a mouse model of metastatic breast cancer” from Panella et al. analyze the role of miR22 in triple-negative breast cancer cells. Since miR22 promoted EMT and invasion in murine models, and its inhibition suppressed the growth of metastases and prolonged the overall survival, the study suggest that miR22 silencing could be a new therapeutic strategy for the treatment of triple-negative breast tumors. The article is esay to read, well written and the results clearly support the conclusions. Just minor changes can improve the manuscript:
1.- Primers for RT-qPCR: it would be nice if the authors show the sequences in a table.
2.- The images from H-E staining and immunohistochemistry should be improved. It is difficult to see anything at their actual size and dark and uneven illumination.
Author Response
We would like to thank the reviewers for their objective and thorough review of our manuscript. Please, find below our detailed response to the reviews. We sincerely believe that we have adequately addressed all the comments raised by the reviewers and, thus, hope that the revised manuscript in its present form could be accepted for publication in Biomedicines.
Responses to Reviewer #1 comments:
The article “Targeting of microRNA-22 suppresses tumor spread in a mouse model of metastatic breast cancer” from Panella et al. analyze the role of miR22 in triple-negative breast cancer cells. Since miR22 promoted EMT and invasion in murine models, and its inhibition suppressed the growth of metastases and prolonged the overall survival, the study suggest that miR22 silencing could be a new therapeutic strategy for the treatment of triple-negative breast tumors. The article is esay to read, well written and the results clearly support the conclusions. Just minor changes can improve the manuscript:
1.- Primers for RT-qPCR: it would be nice if the authors show the sequences in a table.
We thank the Reviewer for the comment and have provided a table with the RT-qPCR primer sequences.
--------------------------------------------------------------------------------------------------------------
2.- The images from H-E staining and immunohistochemistry should be improved. It is difficult to see anything at their actual size and dark and uneven illumination.
We are grateful for this comment, and we addressed it in the reviewed version of figure. All the H&E as well as immunohistochemistry present in figure 3 have been modified to have a better and more even illumination and we made it bigger to help readers in interpreting data

Reviewer 2 Report
The paper is interesting but it must be shortened and partially re-organized.
Title
It should be specified that the paper refers to a mouse model of triple negative metastatic breast cancer
Abstract
This section should better summarize methods and results
Introduction
This section must be shortened.
For example, I suggest to remove the paragraph “In 2018 there were ca. 3.6 million … from one another”.
As the paper is focused on TNBC, I suggest to remove the mention of HER2+ breast cancers.
In addition, some statements are not totally accurate, for example: “HR-/HER2+ and TNBC are the most lethal subtypes, responsible for the vast majority of breast cancer deaths despite comprising less than half of all diagnoses. This disproportion between diagnoses and mortality rates is due to the extreme aggressiveness of these subtypes and the current dearth of treatment options.”
Finally, the Introduction section should not report on results or conclusions.
Methods
As a clinical oncologist, I am not sufficiently skilled in this field.
Results
This section should be shortened.
In particular, the paragraph “We have previously shown that miR-22 contributes to the metastatic potential of breast cancer in an MCF7 xenograft model and a transgenic mouse model overexpressing miR-22. In the xenograft models, miR-22 overexpression markedly increased the prolif-erative rate of MCF7 (Ki67+) cells, as well their metastatic potential to the lung, compared to MCF7 control tumors. Similarly, in transgenic mice engineered to selectively overex-press miR-22 in mammary tissues, we observed a significant decrease in disease-free sur-vival rate resulting from mammary cancer and spontaneous metastatic spread of the dis-ease. These findings prompted us to ask whether miR-22 plays a broader role in breast cancer.
Molecularly, TNBC and HER2+ breast cancers share many similarities as both are enriched in loss-of-function mutations in TP53, PTEN, and RB1. HER2+ breast cancers are further characterized by ERBB2 amplification. These features result in a higher histological grade and a more aggressive disease. In contrast, hormone-receptor positive (HR+) breast cancers typically share a luminal transcriptional profile, which is often estrogen-dependent and enriched in GATA3 mutations with lower mutation rates of TP53” does not report the results of this study and must be moved to another section, perhaps the Discussion section.
The phrase “In terms of prognosis and life expectancy breast cancer is divided into 4 different stages, from I to IV, with stage IV being characterized by metastatic spread and considered in-curable as per today. The vast majority of stage IV breast cancers are also TNBC” is useless and must be removed.
The phrase “Our in-silico analysis suggest that miR-22 levels are elevated in stage IV breast cancer (Fig 1A) and this observation further prompted us to explore the role of miR-22 in this breast cancer subtype” does not report the results of this study and must be moved to another section, perhaps the Introduction section
The results section reports some comments that must be moved to the Discussion section
Discussion
“Although diagnosis of breast cancer cases has risen … to make breast cancer fully curable”.
This paragraphs is useless, I suggest to remove it
“While there are currently several …or the expression of HER2”
Please remove
“In the case of TNBC, treatments are limited to standard chemotherapy, Poly ADP Ribose Polymerase (PARP) inhibitors for tumors harboring BRCA1/2 mutations, and PD-L1 checkpoint blockade for tumors with infiltrating lymphocytes, or tumors that are PD-L1 positive.”
Please check
“Interestingly, antimiR-22 treatment not only impacted the growth of TNBC cells overexpressing miR-22, but was also effective at slowing the growth of TNBC cells expressing basal amounts of miR-22. This finding suggests that an antimiR-22-based therapy could be effective in breast cancer patients irrespective of whether they overexpress miR-22 or exhibit normal expression levels of this miRNA”
Are authors confident about this interpretation?
“Additionally, overexpression of miR-22 dramatically reduced survival of xenograft mouse model based on human TNBC. Given that the mature miR-22 sequence is 100% conserved between mouse and man, our data in mouse models of TNBC are indicative that an antimiR-22-based therapy could be further pursued in patients for the treatment of breast cancer, TNBC and their metastatic spread”
Please modify like this “Additionally, overexpression of miR-22 dramatically reduced survival of xenograft mouse model based on human TNBC. Given that the mature miR-22 sequence is 100% conserved between mouse and man, our data in mouse models of TNBC are indicative that an antimiR-22-based therapy could be further pursued in TNBC patients”
“These findings are critical because they support the notion that the treatment might be beneficial to a broad population of breast cancer patients, and not solely restricted to those overexpressing miR-22, potentially improving survival in breast cancer.”
Please remove.
Author Response
We would like to thank the reviewers for their objective and thorough review of our manuscript. Please, find below our detailed response to the reviews. We sincerely believe that we have adequately addressed all the comments raised by the reviewers and, thus, hope that the revised manuscript in its present form could be accepted for publication in Biomedicines.
Responses to Reviewer #2 comments:
The paper is interesting but it must be shortened and partially re-organized.
Title
It should be specified that the paper refers to a mouse model of triple negative metastatic breast cancer
We thank Reviewer for the comment and have addressed it by adding “triple-negative” to the manuscript title.
--------------------------------------------------------------------------------------------------------------
Abstract
This section should better summarize methods and results
We have edited the Abstract to better reflect the methods used.
--------------------------------------------------------------------------------------------------------------
Introduction
This section must be shortened.
For example, I suggest to remove the paragraph “In 2018 there were ca. 3.6 million … from one another”.
As the paper is focused on TNBC, I suggest to remove the mention of HER2+ breast cancers.
In addition, some statements are not totally accurate, for example: “HR-/HER2+ and TNBC are the most lethal subtypes, responsible for the vast majority of breast cancer deaths despite comprising less than half of all diagnoses. This disproportion between diagnoses and mortality rates is due to the extreme aggressiveness of these subtypes and the current dearth of treatment options.”
Finally, the Introduction section should not report on results or conclusions.
We appreciate the useful comments of the Reviewer and revised the introduction accordingly.
--------------------------------------------------------------------------------------------------------------
Methods
As a clinical oncologist, I am not sufficiently skilled in this field.
--------------------------------------------------------------------------------------------------------------
Results
This section should be shortened.
In particular, the paragraph “We have previously shown that miR-22 contributes to the metastatic potential of breast cancer in an MCF7 xenograft model and a transgenic mouse model overexpressing miR-22. In the xenograft models, miR-22 overexpression markedly increased the prolif-erative rate of MCF7 (Ki67+) cells, as well their metastatic potential to the lung, compared to MCF7 control tumors. Similarly, in transgenic mice engineered to selectively overex-press miR-22 in mammary tissues, we observed a significant decrease in disease-free sur-vival rate resulting from mammary cancer and spontaneous metastatic spread of the dis-ease. These findings prompted us to ask whether miR-22 plays a broader role in breast cancer.
Molecularly, TNBC and HER2+ breast cancers share many similarities as both are enriched in loss-of-function mutations in TP53, PTEN, and RB1. HER2+ breast cancers are further characterized by ERBB2 amplification. These features result in a higher histological grade and a more aggressive disease. In contrast, hormone-receptor positive (HR+) breast cancers typically share a luminal transcriptional profile, which is often estrogen-dependent and enriched in GATA3 mutations with lower mutation rates of TP53” does not report the results of this study and must be moved to another section, perhaps the Discussion section.
The phrase “In terms of prognosis and life expectancy breast cancer is divided into 4 different stages, from I to IV, with stage IV being characterized by metastatic spread and considered in-curable as per today. The vast majority of stage IV breast cancers are also TNBC” is useless and must be removed.
The results section reports some comments that must be moved to the Discussion section
We have addressed all the points raised by the Reviewer in the revised version of the manuscript.
--------------------------------------------------------------------------------------------------------------
The phrase “Our in-silico analysis suggest that miR-22 levels are elevated in stage IV breast cancer (Fig 1A) and this observation further prompted us to explore the role of miR-22 in this breast cancer subtype” does not report the results of this study and must be moved to another section, perhaps the Introduction section
The in silico analysis was carried out using publicly available data bases and since these data have not been published before, we sincerely believe that including these data in the manuscript is warranted.
--------------------------------------------------------------------------------------------------------------
Discussion
“Although diagnosis of breast cancer cases has risen … to make breast cancer fully curable”.
This paragraphs is useless, I suggest to remove it
“While there are currently several …or the expression of HER2”
Please remove
“In the case of TNBC, treatments are limited to standard chemotherapy, Poly ADP Ribose Polymerase (PARP) inhibitors for tumors harboring BRCA1/2 mutations, and PD-L1 checkpoint blockade for tumors with infiltrating lymphocytes, or tumors that are PD-L1 positive.”
Please check
“Additionally, overexpression of miR-22 dramatically reduced survival of xenograft mouse model based on human TNBC. Given that the mature miR-22 sequence is 100% conserved between mouse and man, our data in mouse models of TNBC are indicative that an antimiR-22-based therapy could be further pursued in patients for the treatment of breast cancer, TNBC and their metastatic spread”
Please modify like this “Additionally, overexpression of miR-22 dramatically reduced survival of xenograft mouse model based on human TNBC. Given that the mature miR-22 sequence is 100% conserved between mouse and man, our data in mouse models of TNBC are indicative that an antimiR-22-based therapy could be further pursued in TNBC patients”
“These findings are critical because they support the notion that the treatment might be beneficial to a broad population of breast cancer patients, and not solely restricted to those overexpressing miR-22, potentially improving survival in breast cancer.”
Please remove.
We agree with the Reviewer and have edited the Discussion section accordingly in the revised version of the manuscript.
--------------------------------------------------------------------------------------------------------------
“Interestingly, antimiR-22 treatment not only impacted the growth of TNBC cells overexpressing miR-22,but was also effective at slowing the growth of TNBC cells expressing basal amounts of miR-22. This finding suggests that an antimiR-22-based therapy could be effective in breast cancer patients irrespective of whether they overexpress miR-22 or exhibit normal expression levels of this miRNA”
Are authors confident about this interpretation?
We are confident regarding our interpretation of the data presented in Figure 2 (panel B, D, F and H) and Extended Figure 2 (panel A and C) and Figure 3 (Panel D and F ) on the effect of antimiR-22 inhibition in cells overexpressing the empty control vector, suggesting that the therapeutic approach could be beneficial regardless of the overall levels of miR-22.. However, we have moderated our statement as follows: “Interestingly, antimiR-22 treatment not only impacted the growth of TNBC cells overexpressing miR-22, but affected also the growth of TNBC cells expressing basal amounts of miR-22. This finding suggests that an antimiR-22-based therapy could be effective in breast cancer patients irrespective of whether they overexpress miR-22 or exhibit normal expression levels of this miRNA. However, further studies are required to confirm these observations.

Reviewer 3 Report
In this article, Panella and colleagues have performed straightforward experiments to show that miR-22 inhibition reduces EMT and metastatic and cancer stem markers in 2D and the number/dimension of mammospheres and mouse models of TNBC. The article provides novel insights into miRNA-based cancer therapies, focusing on treating TNBC patients using inhibitory LNA against endogenous miR-22.
The article clearly describes the results to show that miR-22 is a critical player in TNBC aggressiveness.
I have only a few minor points:
1) In the introduction (Version v2), this sentence, lines: 35-47, can be divided into two to make it more straightforward.
2) In the introduction, lines: 68-71: it would be constructive for the readers that do not know anything about miR-22 to write one or two sentences that describe where this miRNA resides in the human genome, that it is evolutionarily conserved if it is intergenic or intragenic.
3) line 93: Please change analysis to analyses.
4) line 200: punctuation is missing.
5) Mice: briefly explain why:
-line: 335 did you choose Nu/J mice?
-line 336: cells injected IV and not orthotopically or IP?
-line 339: the compounds were injected IP and not IV?
Line: 340: I think the dose should be given in mg/kg. Please explain/correct.
6) Figure 2 B, D, E, and C: statistics are missing between LNA-SCR and anti-miR-22 LNA.
Same also for Extended Figure 2
Author Response
Response to reviews of the manuscript by Panella et al. entitled “Targeting of microRNA-22 suppresses tumor spread in a mouse model of triple-negative breast cancer” (biomedicines-2027169).
We would like to thank the reviewers for their objective and thorough review of our manuscript. Please, find below our detailed response to the reviews. We sincerely believe that we have adequately addressed all the comments raised by the reviewers and, thus, hope that the revised manuscript in its present form could be accepted for publication in Biomedicines.
Responses to Reviewer #3 comments:
In this article, Panella and colleagues have performed straightforward experiments to show that miR-22 inhibition reduces EMT and metastatic and cancer stem markers in 2D and the number/dimension of mammospheres and mouse models of TNBC. The article provides novel insights into miRNA-based cancer therapies, focusing on treating TNBC patients using inhibitory LNA against endogenous miR-22.
The article clearly describes the results to show that miR-22 is a critical player in TNBC aggressiveness.
I have only a few minor points:
1) In the introduction (Version v2), this sentence, lines: 35-47, can be divided into two to make it more straightforward.
We thank the Reviewer for the comment and we substituted the mentioned sentence with the following one:
“Among the 5 different breast cancer subtypes triple negative breast cancer (TNBC) is the most aggressive and lethal subtype5. TNBC is characterized by a very strong metastatic potential and its molecular complexity make it particularly hard to treat making the developing for new therapeutic options urgent to address the patient needs.”
2) In the introduction, lines: 68-71: it would be constructive for the readers that do not know anything about miR-22 to write one or two sentences that describe where this miRNA resides in the human genome, that it is evolutionarily conserved if it is intergenic or intragenic.
We are grateful for this comment, and we addressed it in the reviewed version of the manuscript, adding the following sentences:
MicroRNA-22 (miR-22) encoded in exon 2 of the miR-22 host gene MIR22HG, which located on the short arm of Chromosome 17 (GRCh38.p14) in a minimal loss of heterozygosity region. It is highly conserved across many vertebrate species, including chimpanzee, mouse, rat, dog and horse. This level of conservation suggests functional importance.
3) line 93: Please change analysis to analyses.
We have corrected analysis to analyses.
4) line 200: punctuation is missing.
We have edited the text and corrected the missing punctuation
5) Mice: briefly explain why:
-line: 335 did you choose Nu/J mice?
Nu/J mice were used to maximize the chance of engraftment and because they are considered to be an excellent model for xenografting tumor cell lines, aimed to understanding mechanisms of tumor malignancy, and testing the impact of potential treatments (Price, J. E. (2001). "Xenograft models in immunodeficient animals: I. Nude mice: spontaneous and experimental metastasis models." Methods Mol Med 58: 205-213.)
-line 336: cells injected IV and not orthotopically or IP?
We thank the reviewer for this question and have edited the text to make it more clear to the reader. Briefly, we decided to inject MDA-MB-231 cells IV to better mimic the process of metastasis formation and have the cells in the blood stream not to interact with the spreading process or force homing in a specific tissue.
-line 339: the compounds were injected IP and not IV?
Upon optimizing pharmacologic inhibition of miR-22, we tried several administration routes whwe didn’t observe any advantage in IV dosing of the antimiR-22 compound compared to IP. Since IP delivery is much less stressful for animals compared to IV according to the ethics rules listed in the guidelines of Beth Israel Deaconess Medical Center Institutional Animal Care and Use Committee, we decided to administer the antimir-22 compound by IP injection.
Line: 340: I think the dose should be given in mg/kg. Please explain/correct.
We have corrected to dose to 10mg/kg
6) Figure 2 B, D, E, and C: statistics are missing between LNA-SCR and anti-miR-22 LNA.
Same also for Extended Figure 2
We are grateful for this comment and have included the statistics accordingly.

Reviewer 4 Report
In the manuscript entitled “Targeting of microRNA-22 suppresses tumor spread in a mouse model of metastatic triple-negative breast cancer” Riccardo Panella and colleagues investigate the potential of miRNA 22 as an anticancer therapeutic agent. In previous studies the authors had already shown that the overexpression of this miRNA positively impact metastatic potential of breast cancer in an MCF7 xenograft and in a transgenic mouse models. In the present work the pro metastatic potential of miR-22 was confirmed in triple negative breast cancer (TNBC) models. Moreover, a reduction of EMT in in-vitro and in-vivo was obtained by miR-22 inhibition. Interestingly, miR-22 inhibition takes effect not only in TNBC cells that overexpress miR-22 but also in TNBC cells with basal miR-22 amounts, suggesting the possibility to develop new miR-22 based therapeutic strategies. The manuscript is interesting, the research design is appropriate, the results well presented and opportunely discussed.
Minor comments
Line 77-83 of results should be part of the introduction (previous results)
Line 183-194 of discussion are a repeat and should be removed
Author Response
Response to reviews of the manuscript by Panella et al. entitled “Targeting of microRNA-22 suppresses tumor spread in a mouse model of triple-negative breast cancer” (biomedicines-2027169).
We would like to thank the reviewers for their objective and thorough review of our manuscript. Please, find below our detailed response to the reviews. We sincerely believe that we have adequately addressed all the comments raised by the reviewers and, thus, hope that the revised manuscript in its present form could be accepted for publication in Biomedicines.
Responses to Reviewer #4 comments:
In the manuscript entitled “Targeting of microRNA-22 suppresses tumor spread in a mouse model of metastatic triple-negative breast cancer” Riccardo Panella and colleagues investigate the potential of miRNA 22 as an anticancer therapeutic agent. In previous studies the authors had already shown that the overexpression of this miRNA positively impact metastatic potential of breast cancer in an MCF7 xenograft and in a transgenic mouse models. In the present work the pro metastatic potential of miR-22 was confirmed in triple negative breast cancer (TNBC) models. Moreover, a reduction of EMT in in-vitro and in-vivo was obtained by miR-22 inhibition. Interestingly, miR-22 inhibition takes effect not only in TNBC cells that overexpress miR-22 but also in TNBC cells with basal miR-22 amounts, suggesting the possibility to develop new miR-22 based therapeutic strategies. The manuscript is interesting, the research design is appropriate, the results well presented and opportunely discussed.
Minor comments
Line 77-83 of results should be part of the introduction (previous results)
We are thankful for this comment and have moved the indicated section to Introduction.
Line 183-194 of discussion are a repeat and should be removed
We have removed the redundant lines from the Discussion.

Round 2
Reviewer 2 Report
Authors have only minimally modified the text according my suggestion. The paper cannot be published in the present form
Author Response
We are sorry that our answer were considered not satisfactory to this reviewer.